# Carbohydrate Counting, Empowerment and Glycemic Outcomes in Adolescents and Young Adults with Long Duration of Type 1 Diabetes

**DOI:** 10.3390/nu15224825

**Published:** 2023-11-18

**Authors:** Elisabeth Jelleryd, Anna Lena Brorsson, Carmel E. Smart, Ulrika Käck, Anna Lindholm Olinder

**Affiliations:** 1Department of Clinical Science and Education, Södersjukhuset, Karolinska Institute, 11883 Stockholm, Sweden; ulrika.kack@regionstockholm.se (U.K.); anna.lindholm.olinder@ki.se (A.L.O.); 2Women’s Health and Allied Health Professionals Theme, Medical Unit Clinical Nutrition, Karolinska University Hospital, 17176 Stockholm, Sweden; 3Department of Neurobiology, Care Sciences and Society, Karolinska Institute, 14152 Stockholm, Sweden; anna-lena.brorsson@ki.se; 4Department of Endocrinology, John Hunter Children’s Hospital, Newcastle, NSW 2305, Australia; carmel.smart@health.nsw.gov.au; 5School of Health Sciences, University of Newcastle, Newcastle, NSW 2300, Australia; 6Sachs’ Children and Youth Hospital, Södersjukhuset, 11883 Stockholm, Sweden

**Keywords:** adolescents, carbohydrate counting, children, diabetes type 1, empowerment, young adults

## Abstract

The complex treatment for diabetes type 1 (T1D) includes insulin dosing for every meal, which requires education and experience to achieve optimal outcomes. Advanced carbohydrate counting (ACC) is the recommended method. We studied ACC as part of a standard treatment with the aim to explore its associations with glycemic control and empowerment in adolescents and young adults. We used national registry data on glycemic outcomes, a study-specific questionnaire regarding the use of ACC and the Gothenburg Young Persons Empowerment Scale (GYPES) to measure empowerment. A total of 111 participants (10–28 years of age, diabetes duration >9 years, mean HbA1c of 55.4 mmol/mol) answered the questionnaire. We found that most participants (79.3%) who learn ACC, at onset or later, continue to use the method. A higher level of empowerment was associated with lower HbA1c (*p* = 0.021), making patient empowerment an important factor in achieving optimal glycemic outcomes. No associations were found between ACC and empowerment or glycemic outcomes. A mixed strategy, only using ACC sometimes when insulin dosing for meals, was associated with the lowest empowerment score and highest HbA1c and should warrant extra education and support from the diabetes team to reinforce a dosing strategy.

## 1. Introduction

Type 1 diabetes (T1D) is a lifelong disease with a multifactorial treatment, which requires exogenous insulin therapy, calculated insulin doses for every meal and constant attention to blood glucose levels. The recommended method for calculating insulin for meals is carbohydrate counting, a method based on carbohydrates being the main factor affecting the postprandial glycemic response [1]. Calculating a bolus dose of insulin using advanced carbohydrate counting (ACC) involves knowing the carbohydrate content in food in grams divided by an insulin to carbohydrate ratio (I:C) and a correction dose based on a sensitivity factor (ISF) to lower high glucose values [2].

There are different methods of quantifying carbohydrates, aside from counting in 1-g increments; also, 10-g portions and 15-g exchanges are used and one has not been proven to be more advantageous than the other. To achieve improvements in glycemic outcomes, it seems to be the insulin dose adjustment to changes in food quantity that is most important, rather than how you quantify the carbohydrates [3]. Research has shown that an insulin dose may cover a range of carbohydrates [4], and when investigating accuracy of carbohydrate estimation in children and their caregivers, they were within that range [3]. Furthermore, adjustments to insulin doses as well as patterns of delivery are needed according to the fat and protein content of meals [1].

Overall, diabetes self-management is burdensome and requires experience and continuous education to achieve optimal glycemic outcomes [5]. Children with T1D are dependent on their parents and adults in their immediate surroundings to care for their diabetes and insulin dosing for meals. As children grow and gradually reach autonomy, the responsibility for the diabetes self-management is transferred from the parent to the adolescent. This is a critical period in life and teenagers are the least likely to meet glycemic targets of all pediatric age groups [6]. Finding tools and strategies to help adolescents maintain carbohydrate counting skills and acceptable glycemic outcomes is of great importance.

Carbohydrate counting has been used in diabetes treatment since the 1990s and is considered to help achieve glycemic control, while at the same time allowing flexibility in food choices. These dual qualities are desirable when working with children and adolescents [7]. Guidelines recommend ACC from the onset of diabetes, although this has never been studied and is based on clinical experience [1]. Most studies on ACC have been intervention studies, examining its glycemic lowering qualities, but results are conflicting [8,9]. The long-term effect of ACC on glycemic outcomes has not been studied.

Factors found to be associated with glycemic outcomes in T1D are greater dietary knowledge, higher self-efficacy and empowerment [10]. Patient empowerment is considered to be “An enabling process or outcome arising from communication with the healthcare professional and a mutual sharing of resources over information relating to illness, which enhances the patient**’**s feelings of control, self-efficacy, coping abilities and ability to achieve change over their condition” [11] (p. 2). The association between empowerment and ACC has not been studied.

Our hypothesis was that individual empowerment for diabetes self-management is associated with positive glycemic outcomes and the use of advanced carbohydrate counting increases the feeling of empowerment.

The aim of this study was to explore the relationship between advanced carbohydrate counting, empowerment and glycemic outcomes in adolescents and young adults with T1D.

## 2. Materials and Methods

### 2.1. Study Participants

People with onset of type 1 diabetes between 2010 and 2013 (*n* = 364) at Astrid Lindgren Children’s Hospital, Stockholm, Sweden, were invited to participate in the study. This period was chosen because ACC was introduced as part of the standard treatment method in January of 2012 for all newly diagnosed children with T1D at the two pediatric diabetes units at Astrid Lindgren Children’s Hospital. This allowed comparisons between those who were given fixed insulin doses at onset, prescribed by doctors to suit standard meals throughout the day and not based on an insulin-to-carbohydrate ratio (2010–2011), with those who were taught ACC from onset (2012–2013). A total of 111 responses were obtained (30.5%), ranging from 10 to 28 years old, and 34 (30.6%) were <18 years old. Characteristics of participants are shown in Table 1.

Power to measure differences in glycemic outcomes between carbohydrate counters and non-carbohydrate counters were calculated based on HbA1c-values. To detect a difference of 6 mmol/mol (0.6% DCCT) in HbA1c (SD: ±9 mmol/mol (±0.91%-units DCCT), based on values from the National Diabetes Registry), at least 74 participants should be recruited, with 37 in each group (power 80% and alpha 0.05).

### 2.2. Questionnaires

A letter with a study invitation along with a consent form was sent in June of 2022 to all potential participants. Reminders were sent on two additional occasions, in August and October of 2022.

Two digital questionnaires were used, a study-specific questionnaire to ask questions regarding the use of ACC and the Gothenburg Young Person Empowerment Scale (GYPES) to measure diabetes specific empowerment [12]. Participants could access the questionnaires by scanning a QR code received in the invitation letter and entering their unique study ID. Participants aged 12 or younger were advised to fill out the questionnaires with a guardian.

The study-specific questionnaire consisted of a combination of 22 short-ended, multiple-choice and open-ended questions regarding the use of ACC and to what extent it was used. Participants could mark using ACC for “all meals”, for “most meals”, using it “sometimes” and “not using ACC” when insulin dosing for meals. Participants were grouped based on answers into three groups. If ACC was used for all or most meals, they were considered to use ACC as a dosing strategy for meals. If they had answered using ACC sometimes, they were considered to use a mixed strategy, and those who marked not using ACC were considered as using other strategies for insulin dosing. Additionally, questions regarded the participants’ social situation, the education level of their parents and if they were living alone. Questions on food restriction related to their diabetes were also asked including to what extent they intentionally limited their carbohydrate intake, if they thought about how much carbohydrate they should consume and if they avoided or limited foods, meals or meal situations such as restaurant visits.

The Gothenburg Young Persons Empowerment Scale is a 15-item validated questionnaire measuring empowerment in young persons with chronic conditions. The participants rate their answers on a five-grade Likert scale [12]. The questions are divided into 5 domains: knowledge and understanding, personal control, identity, shared decision making and enabling others. Due to an error when making the digital questionnaire, a question was mistakenly excluded from the domain of personal control and scores were therefore calculated using 2 questions instead of 3. The scores were multiplied by 1.5 to agree with the questionnaires total score, with a maximal score of 75 points. This has not significantly impacted the significance of the results, as answers were homogenous for each domain.

### 2.3. Registry Data

Data from the National Diabetes Registry (NDR) in Sweden was applied for and obtained in April 2022 for all participants (*n* = 364). Variables included HbA1c, mean blood glucose (mean BG), time in range defined as 3.9–10 mmol/L (TIR) and time in target defined as 3.9–7.8 mmol/L (TIT), standard deviation (SD), as well as pump or pen treatment and use of continuous glucose monitor (CGM). In NDR, TIR is reported for adults and TIT is reported for children. To allow comparisons for both groups, TIT for children were calculated into TIR using the formula TIT × 1.4 [13].

In the present study, only registry data from the participants that answered the questionnaires were analyzed (*n* = 111).

### 2.4. Ethics

The study was conducted in accordance with the Declaration of Helsinki and approved by the Swedish Ethical Review Authority, protocol code 2021-04903, on 11 January 2022. The study is registered in the Karolinska University Hospital database for clinical studies, K 2021-7905.

All participants were sent study information and informed consent and agreed to participate by accepting informed consent digitally in the questionnaire.

### 2.5. Statistical Analysis

Statistical analyses were conducted using IBM SPSS Statistics version 28.0.1.1. Significance level was set at *p* < 0.05. Chi-square and one-way ANOVA was used to compare groups of participants and glycemic control. Test of normality was performed which showed groups were not normally distributed for result analysis of empowerment scores; therefore, non-parametric independent-samples Kruskal –Wallis tests were performed.

## 3. Results

### 3.1. Study Population

Fixed insulin doses from the onset of diabetes were taught to 47.7% of the participants and 52.3% were taught ACC. There were no statistical differences between the two groups in age, age at onset, sex, multiple daily injections (MDIs)/insulin pump treatments (IPTs) or what insulin dosing strategy they used now. The group that had been taught fixed doses from onset had significantly longer diabetes duration (*p* < 0.001), as expected (Table 1).

There was no significant difference between groups based on method learnt at onset apart from diabetes duration (*p* < 0.001). When categorized based on weight, there was no difference in number of overweight or obese participants in the different groups.

In the group that was taught fixed doses from onset, 41 participants (77%) had learnt and used ACC. In the group who had been taught ACC from onset, 47 participants (81%) continued to use ACC. Independent of method taught at onset, 88 participants (79.3%) were using ACC as a strategy when insulin dosing for all or most meals. In the ACC group, 11 (19%) had stopped using the strategy or used a mixed strategy. Reasons for choosing to stop were related to ACC being too time consuming and that they had learnt how much insulin to dose without calculating carbohydrates. The group who did not use ACC marked using experience and ‘knowing’ what to dose for meals. A low carbohydrate diet was used by 2 participants instead of ACC, and these participants were both older than 20 years old.

We found no differences between the groups in response to food or carbohydrate limitations or avoidance.

### 3.2. Advanced Carbohydrate Counting and Glycemic Control

Participants were categorized according to current insulin dosing strategy and compared, displayed in Table 2. Participants in the group who used ACC strategy were significantly younger (*p* = 0.001) and were younger at diabetes onset than participants in the group who used other strategies, which had the highest mean age (*p* = 0.006). There was a significantly higher use of insulin pumps in the ACC group (*p* < 0.001).

There was a difference in mean BG between the groups (*p* = 0.009). The group using mixed strategy had the highest mean BG and the group using other strategies had the lowest. However, there were no statistically significant differences in HbA1c, time in range and glucose variability (SD) across groups.

### 3.3. Advanced Carbohydrate Counting and Empowerment

We found no differences in feeling of empowerment between groups using different dosing strategies, displayed in Table 3 and Figure 1. However, we found there was a difference in empowerment associated with glycemic control (Figure 1). The higher the total score of empowerment, the lower the HbA1c (*p* = 0.021).

## 4. Discussion

In this study, we investigated the relationship between advanced carbohydrate counting, glycemic outcomes and empowerment in 111 participants with a long duration of T1D. We could confirm our hypothesis that a higher level of empowerment is associated with better glycemic outcomes; however, we did not find an association between the use of ACC and empowerment or ACC and HbA1c.

It is not established in what direction the positive association lies between empowerment and glycemic control, and whether a higher feeling of empowerment results in a lower HbA1c or if a lower HbA1c gives a higher feeling of empowerment. However, our findings emphasize that empowerment is important in diabetes self-management to achieve better glycemic outcomes. In a review of factors affecting glycemic control, Cheng et al. wrote that “higher empowerment level enhances self-management practices” and concluded that empowerment, together with socio economic status and nutrition knowledge, as well as self-efficacy, were key elements to improve glycemic outcomes [10].

To the best of our knowledge, this is the first study to examine the association of ACC as standard treatment from diabetes onset on glycemic outcomes. We found no statistically significant association between ACC and improved glycemic outcomes or empowerment; however, our data showed that most participants who were taught carbohydrate counting continued to use the strategy with an overall satisfying glycemic control with a mean HbA1c of 54.6 mmol/mol (7.2%). Furthermore, we found that participants who were not taught ACC from onset had adapted the method and used it to a similar extent to those who were taught ACC from onset, achieving equal blood glucose outcomes, indicating that it is never too late to learn to carbohydrate count. However, ACC was adopted as the standard treatment in 2012 and all existing patients in our clinic who had not been taught ACC from diabetes onset were encouraged to start. This suggests that it is important to encourage patients to not only learn to use ACC but to adhere to it through unified messaging and endorsement of the method from the diabetes team.

Previous studies, examining the effectiveness of ACC and its HbA1c-lowering qualities have been interventions and the results have been conflicting. ACC has shown to lower HbA1c mainly in adults with a high baseline HbA1c [2,8,9]. The follow-up time across studies has varied, with only a few studies longer than 12 months, and the positive effect of ACC on HbA1c has decreased over time. Our data support that ACC as standard treatment, taught both from diabetes onset and later, continues to be well used, resulting in near-target glycemic outcomes after ≥9 years of diabetes duration. ACC can therefore be considered a user-friendly tool which can be helpful in achieving optimal glycemic outcomes.

The group using “Mixed strategy” had the poorest glycemic outcomes, which is of clinical importance. This group had significantly higher mean BG. They also had higher HbA1c, lower TIR and larger variability compared to the other groups; however, these differences were not statistically significant. We believe this reflects a lack of clear strategy when insulin dosing for meals. They marked using ACC sometimes in the questionnaire, which could imply a lack of knowledge in any of the parts necessary for using ACC and perhaps lack of experiential learning, as they also had shorter mean diabetes duration [14,15]. The group had a mean age of 19.4 and was therefore comprised mainly of teenagers, a period known to be turbulent and with the poorest glycemic control in the pediatric cohort [6]. In a qualitative study exploring teenagers experience and use of carbohydrate counting, the method was described as simple and useful with a positive influence on blood glucose but that they “didn’t bother to use it” [15] (p. 892). Finding out the carbohydrate content of certain foods was a hindrance to using ACC and the authors suggested individualized follow ups and training to dose insulin adapted to each individuals’ capacity and life situation [15]. The lack of strategy should therefore raise concern in the diabetes team to be targeted for education and support.

Other possible reasons for the inferior outcomes could be that young adults who have recently transitioned to adult care have been shown to further increase their HbA1c, according to National Registry Data [6]. This could possibly be explained, in part, by the change in glycemic targets in Swedish diabetes care when adolescents are transferred to adult care, where HbA1c goals increase to 52 mmol/mol from 48 mmol/mol and a goal of 50% TIT increases to 70% when TIR is used [6]. Lower targets for HbA1c levels have shown to affect a center’s outcome [16,17].

Our data among participants who used ACC strategy show that IPT was the predominant mode of therapy, suggesting that insulin pumps require the use of ACC. Qualitative research has shown ACC is facilitated by the pump; teenagers on IPT described the pumps to be doing “most of the work for them”, as the teenagers added carbohydrates but the pump calculated the insulin dose [15] (p. 892). The majority of participants in the “Mixed strategy” group were on MDI and therefore did not have the advantage of using a pump bolus calculator. Therefore, without a bolus calculator and sufficient experiential learning, ACC might be used less resulting in worsened glycemic outcomes. Teaching methods for estimation and insulin dose calculation for people on MDI remains a challenge. 

The group “Other Strategy”, who did not use ACC, were experiential learners and had the lowest mean BG as well as the highest TIR, though the latter was not significantly different from the other groups. They were also significantly older and had the longest diabetes duration. This is in line with previous observations that the probability for reaching glycemic targets increases with age as an adult [18].

Response rate of the questionnaires was sufficient based on initial power calculations to detect differences in HbA1c between groups; however, the uneven distribution in the groups as well as tight glycemic control was a limitation for detecting differences across groups. Nonetheless, the groups are likely to be representative of the population as ACC is well incorporated and used by the majority of children, adolescents and young adults with T1D in Stockholm, Sweden.

The questionnaire used to determine if ACC was used was not validated, which could be considered a limitation in the study. When cross-referencing questionnaire answers to reported NDR data (carbohydrate counting yes or no), they were well matched for those whom a value existed. However, the gradation is lost when reporting is only Yes or No and therefore the data from the questionnaire offered nuanced information, better suited for answering the aim of this study.

The study population had high access to diabetes technology, where use of continuous glucose monitors was 83.6% and insulin pump use was 68.1%. This is consistent with data for the Swedish population for the age groups investigated [6] but not necessarily other populations where access is dependent on economic means and insurance coverage, hence, limiting generalizability.

The main strength of this study is that we have been able to evaluate the use of ACC in a clinical setting a decade after diabetes onset and compare individuals who were taught ACC from the start to those who were taught ACC at a later occasion. We demonstrated that ACC is an insulin dosing strategy that is well used after long durations of T1D, which is an important contribution to the knowledge of ACC. Nevertheless, more research is needed to understand how to empower individuals living with T1D and personalize meal carbohydrate estimations and dosing strategies.

In summary, the results are in line with previous findings [2,8] that the efficacy of ACC is not well established across all forms of insulin therapy, nor completely understood, but continues to be the method of choice in current guidelines [1]. Our results are of clinical relevance as they show the benefit of being taught ACC both from diabetes onset and later. The lack of a dosing strategy should warrant extra support from the diabetes team, as this group has challenges in achieving target glycemia.

## 5. Conclusions

Only using ACC sometimes when insulin dosing for meals resulted in the lowest empowerment score and highest HbA1c. Patient empowerment is an important factor in achieving optimal glycemic outcomes. Advanced carbohydrate counting as the standard treatment method contributes to satisfying glycemic outcomes. The method is well used and accepted, and should be taught at onset, in consistency with current guidelines; however, it is never too late to start.

## Figures and Tables

**Figure 1 nutrients-15-04825-f001:**
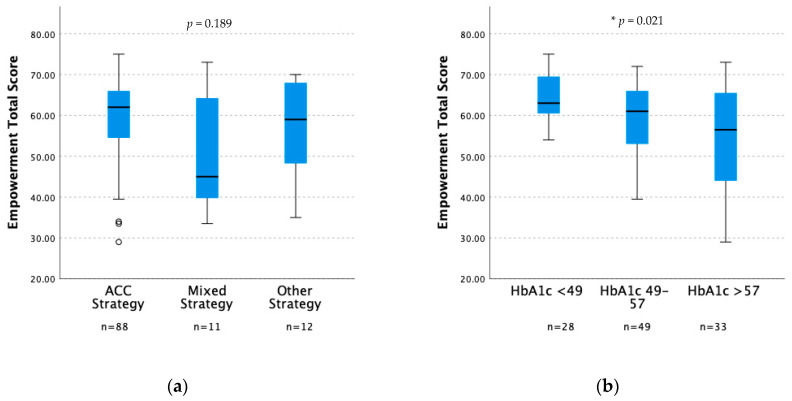
Box plots to visualize empowerment, total score according to different strategies for insulin dosing (ACC, mixed, other) and different levels of HbA1c. * indicating significant difference. Independent-sample Kruskal–Wallis test was performed. (**a**) Empowerment, total score (*y*-axis), in groups using different insulin dosing strategies, *p* = 0.189. (**b**) Empowerment, total score (*y*-axis), in groups with different glycemic control, *p* = 0.021.

**Table 1 nutrients-15-04825-t001:** Characteristics of participants with diabetes onset in 2010–2011 who were taught standard doses from onset compared to participants with diabetes onset in 2012–2013 who were taught ACC from onset.

		Fixed Insulin Dose from Onset *n* = 53	ACC from Onset *n* = 58	*p*-Value
Age (mean)		18.81 ± 4.76	17.93 ± 4.86	0.338
Age at onset		8.4 ± 4.65	9.4 ± 4.82	0.247
Diabetes duration (SD)		11.25 (0.61)	9.34 (0.55)	<0.001 *
Male/Female		25/28	29/29	0.766
Weight ** (*n* = 95)	Normal weight	31 (69%)	37 (74%)	0.681
	Overweight	13 (29%)	11 (22%)	
	Obese	1 (2%)	2 (4%)	
HbA1c mean last year (mmol/mol) (SD) (*n* = 110)		55.7 (12.8)	55.0 (10.8)	0.754
HbA1c mean last year (% DCCT) (SD) (*n* = 110)		7.25 (0.6)	7.18 (0.6)	
Pen/Pump		16/37	20/38	0.629
ACC strategy		41 (77%)	47 (81%)	0.065
Mixed strategy		3 (6%)	8 (14%)	
Other strategy (no ACC)		9 (17%)	3 (5%)	

** Weight categorized according to BMI; <18.5 kg/m^2^ underweight (no cases), 18.5–24.9 kg/m^2^ normal weight, 25.0–29.9 kg/m^2^ overweight, >30.0 kg/m^2^ obesity. * *p* < 0.05 denotes significantly different. One-way ANOVA and Chi-square tests were performed.

**Table 2 nutrients-15-04825-t002:** Characteristics of participants divided into groups by choice of insulin dosing strategy.

		ACC Strategy (*n* = 88)	Mixed Strategy (*n* = 11)	Other Strategy (*n* = 12)	*p*-Value
Age	Mean	17.6	19.4	22.7	0.001 *
	CI	16.6, 18.6	16.3, 22.4	20.3, 25.2	
Age at onset	Mean	8.3	10.3	12.7	0.006 *
	CI	7.3, 9.3	7.1, 13.5	10.2, 15.1	
Diabetes duration	Mean	10.1	9.6	11.0	0.027 *
	CI	9.8,10.4	9.1, 10.6	10.3, 11.7	
Pen / Pump	n	21/67	7/4	8/4	<0.001 *
	%	23.9/76.1	63.6/36.4	66.7/33.3	
Hba1c last year IFCC (mmol/mol) (*n* = 110)	Mean	54.6	60.4	56.5	0.281
	CI	52.9, 57.0	50.7, 70.2	50.8, 62.1	
HbA1c last year DCCT (%) (*n* = 110)	Mean	7.15	7.7	7.3	
	CI	7.0, 9.0	6.8, 8.6	6.8, 7.8	
Mean blood glucose (mmol/L) (*n* = 92)	Mean	9.0	10.3	8.4	0.009 *
	CI	8.7, 9.3	8.5, 12.2	7.3, 9.6	
Time in range (TIR) last year % (*n* = 88)	Mean	60.7	48.9	63.2	0.085
	CI	57.2, 64.1	32.7, 65.1	48.6, 77.8	
Standard deviation (SD) (*n* = 82)	Mean	3.4	4.1	3.3	0.078
	CI	3.2, 3.6	3.2, 4.9	2.7, 3.9	

* *p* < 0.05 denotes significantly different. One-way ANOVA test was used.

**Table 3 nutrients-15-04825-t003:** Empowerment scores, total and by subgroups, in the different groups arranged by use of insulin dosing strategy.

Empowerment	ACC Strategy (*n* = 88)		Mixed Strategy (*n* = 11)		Other Strategy (*n* = 12)		*p*-Value
	Median	Min Max	Median	Min Max	Median	Min Max	
Total score	62	29, 75	45	33.5, 73	59	35, 70	0.189
Knowledge	14	5, 15	11	7, 15	15	7, 15	0.104
Personal control	13.5	6, 15	12	6, 15	14.2	6, 15	0.408
Identity	12	4, 15	10	6, 14	11	5, 13	0.186
Shared decision making	12	4, 15	12	6, 15	12	6, 15	0.663
Enabling others	11.5	3, 15	8	3, 15	9	3, 15	0.076

## Data Availability

The data presented in this study are available on request from the corresponding author. The data are not publicly available due to ethical reasons.

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
