# Peer review of "Carbohydrate Counting, Empowerment and Glycemic Outcomes in Adolescents and Young Adults with Long Duration of Type 1 Diabetes"

_nutrients, 2023, doi:10.3390/nu15224825_

Round 1
Reviewer 1 Report
Comments and Suggestions for Authors
Very interesting study on the association between empowerment and advance3d carbohydrat counting.
This are my observations:
1) Line 49: “Furthermore, adjustments of insulin dose should also be made to the fat and protein con-48 tent of foods to further optimize glucose outcomes”: that's not entirely correct. There is no clear evidence of the effectiveness of protein lipid formula compared to carbohydrate counting. Please clarify this concept better using the same reference you cited (ISPAD 20222).
2) I think that you need to better clarify, also in the abstract, that the results of the study are in line with other studies already present in the literature (time in range, HbA1c). Above all that it is not essential that patient it trained on ACC (since the diabetes onset). Mixed strategies and basic carb counting are enough to achieve good glycemic results (especially in pump users) but that even a mixed or basic strategy can achieve the same result on metabolic control.
3) I think that a weakness of the study was that the BMI (zscore) was not evaluated .It would be useful to add this parameter in the description of the participants (table 1). I believe that the comparison with this parameter would have enriched the results and made them more complete
Author Response
Thank you very much for taking the time to review this manuscript and for your comments. Please find the detailed responses below and the corresponding revisions as well as in track changes in the re-submitted file.
- Line 49: “Furthermore, adjustments of insulin dose should also be made to the fat and protein con-48 tent of foods to further optimize glucose outcomes”: that's not entirely correct. There is no clear evidence of the effectiveness of protein lipid formula compared to carbohydrate counting. Please clarify this concept better using the same reference you cited (ISPAD 20222).
Thank you for pointing this out, we have made changes according to your suggestion.
Furthermore, adjustments to insulin doses as well as patterns of delivery are needed according to the fat and protein content of meals.
- I think that you need to better clarify, also in the abstract, that the results of the study are in line with other studies already present in the literature (time in range, HbA1c). Above all that it is not essential that patient it trained on ACC (since the diabetes onset). Mixed strategies and basic carb counting are enough to achieve good glycemic results (especially in pump users) but that even a mixed or basic strategy can achieve the same result on metabolic control.
Thank you for bringing attention to clarifying the reporting of data. I hope you will find we have made sufficient changes. We have added to the summary that our research is in line with previous findings, and we have made changes to clarify our findings in the abstract.
As regard to the effect of mixed strategy we would like to clarify those results. Our data showed, that although it was a not statistically significantly, there were clinically significant differences as HbA1c was 60.4 mmol/mol (7.7%) in the mixed strategy group as compared to the ‘ACC’ and ‘Other’ group which had 54.6 (7.15%) and 56.5 mmol/mol (7.3%) respectively. The paragraph discussing the results from the Mixed strategy group has been restructured to offer better clarity (as per track changes in manuscript).
- I think that a weakness of the study was that the BMI (zscore) was not evaluated. It would be useful to add this parameter in the description of the participants (table 1). I believe that the comparison with this parameter would have enriched the results and made them more complete.
We have added BMI-data to table 1 categorized as normal weight, overweight and obese. As suggested, this offers a nice addition to the description of participants.
Weight** (n=95) |
Normal weight |
31 (69%) |
37 (74%) |
0.681 |
Overweight |
13 (29%) |
11 (22%) |
||
Obese |
1 (2%) |
2 (4%) |

Reviewer 2 Report
Comments and Suggestions for Authors
Based on the hypothesis that individual empowerment in diabetes self-management is associated with positive glycaemic outcomes and that the use of advanced carbohydrate counting increases feelings of empowerment, the article aims to explore the relationship between advanced carbohydrate counting, empowerment and glycaemic outcomes in adolescents and young adults with type 1 DM. To do so, the authors designed an experimental study involving 111 adolescents and young adults aged 10-28 years, all patients at the Astrid Lindgren Children's Hospital in Stockholm, Sweden, who used different carbohydrate counting strategies.
The work is particularly interesting. On the one hand, it concerns a growing patient population, those with diabetes. On the other hand, it is of particular relevance, when diabetes is diagnosed in early life and adolescence, to monitor carbohydrate intake in order to determine the exact insulin dose. Since advanced carbohydrate counting has been shown to be an effective method for insulin control, it is of interest that children and young people are trained in its use and management from an early stage.
It should be noted that the design of the study to test the hypothesis is appropriate, as well as the instruments used and the data processing carried out, as they allow the relationship between the parameters in the different groups of participants to be evidenced. Likewise, the recording of the results in the tables included in the article is clear and explicit, therefore, the presentation of the results is adequate and understandable.
As for the discussion, it is adequate on the basis of the results obtained in the research, commenting on the results achieved and contrasting them with those obtained by other authors, although not too many studies have been used for this purpose. The authors themselves comment on this limitation as well as the fact that the questionnaire to determine whether advanced carbohydrate counting is used has not been validated. However, the evaluation of the use of advanced carbohydrate counting is clearly stated and that this is an insulin dosing strategy that is well used after a long duration of DM1.
This article emphasises the need for further research on the need to empower individuals living with DM1 to personalise mealtime carbohydrate estimation and dosing strategies.
Finally, I also consider the conclusions drawn from the study to be appropriate.
Author Response
We thank reviewer 2 for reviewing this manuscript and their feedback.